# Service user experiences and views regarding telemental health during the COVID-19 pandemic: A co-produced framework analysis

**Norha Vera San Juan**[1]*, **Prisha Shah**[2], **Merle Schlief**[3], **Rebecca Appleton**[3], **Patrick Nyikavaranda**[2], **Mary Birken**[3], **Una Foye**[1], **Brynmor Lloyd-Evans**[3], **Nicola Morant**[3], **Justin J. Needle**[4], **Alan Simpson**[1], **Natasha Lyons**[3], **Luke Sheridan Rains**[3], **Zainab Dedat**[3], **Sonia Johnson**[3]

**1** Department of Health Service and Population Research, NIHR Mental Health Policy Research Unit, King's College London, London, United Kingdom, **2** NIHR Mental Health Policy Research Unit COVID-19 Co-Production Group, London, United Kingdom, **3** Division of Psychiatry, NIHR Mental Health Policy Research Unit, University College London, London, United Kingdom, **4** Division of Health Services Research and Management, City University of London, London, United Kingdom

* norha.vera@kcl.ac.uk

**Data Availability Statement:** Full transcriptions of the interviews analyzed in this study cannot be shared publicly to protect the privacy and

## Abstract

### Background

The prominence of telemental health, including providing care by video call and telephone, has greatly increased during the COVID-19 pandemic. However, there are clear variations in uptake and acceptability, and concerns that digital exclusion may exacerbate previous inequalities in access to good quality care. Greater understanding is needed of how service users experience telemental health, and what determines whether they engage and find it acceptable.

### Methods

We conducted a collaborative framework analysis of data from semi-structured interviews with a sample of people already experiencing mental health problems prior to the pandemic. Data relevant to participants' experiences and views regarding telemental health during the pandemic were identified and extracted. Data collection and analysis used a participatory, coproduction approach where researchers with relevant lived experience, contributed to all stages of data collection, analysis and interpretation of findings alongside clinical and academic researchers.

### Findings

The experiences and preferences regarding telemental health care of the forty-four participants were dynamic and varied across time and settings, as well as between individuals. Participants' preferences were shaped by reasons for contacting services, their relationship with care providers, and both parties' access to technology and their individual preferences.

While face-to-face care tended to be the preferred option, participants identified benefits of remote care including making care more accessible for some populations and improved

anonymity of participants. If you wish to obtain access to this data, please contact the UCL ethics committee on ethics@ucl.ac.uk and/or the corresponding author.

**Funding:** This paper presents independent research commissioned and funded by the National Institute for Health Research (NIHR) Policy Research Programme, conducted by the NIHR Policy Research Unit (PRU) in Mental Health. https://www.nihr.ac.uk/explore-nihr/funding-programmes/policy-research.htm The views expressed are those of the authors and not necessarily those of the NIHR, the Department of Health and social care, or its arm's length bodies or other government departments

**Competing interests:** The authors have declared that no competing interests exist.

efficiency for functional appointments such as prescription reviews. Participants highlighted important challenges related to safety and privacy in online settings, and gave examples of good remote care strategies they had experienced, including services scheduling regular phone calls and developing guidelines about how to access remote care tools.

## Discussion

Participants in our study have highlighted advantages of telemental health care, as well as significant limitations that risk hindering mental health support and exacerbate inequalities in access to services. Some of these limitations are seen as potentially removable, for example through staff training or better digital access for staff or service users. Others indicate a need to maintain traditional face-to-face contact at least for some appointments. There is a clear need for care to be flexible and individualised to service user circumstances and preferences. Further research is needed on ways of minimising digital exclusion and of supporting staff in making effective and collaborative use of relevant technologies.

## Introduction

Telehealth has been a primary approach internationally for maintaining health care during the COVID-19 pandemic [1, 2]. The UK implemented personal distancing measures on the 16th of March, 2020, and National Health Service (NHS) mental health care providers responded with extensive adoption of telemental health, including video calls and phone calls, to maintain service delivery in conditions of lockdown, and then subsequently with social distancing measures in place. Telemental health adoption was somewhat piecemeal, with policies and platforms varying between services and provider Trusts, but most community services shifted to a majority of patient contacts being remote, but with some face-to-face contact continued where deemed essential, especially in crisis care settings (7) Huge increases in remote patient consultations have subsequently been seen in NHS primary care and mental health settings with a surge in patients' uptake of remote health services, including registrations for the NHS App, NHS login and e-prescription services [3, 4]. As the pandemic continues, mental health services have faced severe disruption, with increased demand alongside reductions in capacity and infection control requirements [5, 6]. It is in this context that the rapid and widespread adoption of remote delivery of mental health services (telemental health) has been central to maintaining assessment, treatment and support in community, hospital and residential settings [2, 7].

Telehealth has been defined as "the delivery of health-related services and information via telecommunications technologies in the support of patient care, administrative activities, and health education" [8]. Before the pandemic, numerous research studies found evidence of telemental health effectiveness in reducing treatment gaps and improving access to care across a wide range of populations, settings and conditions [9–15]. Overall, the evidence suggests that synchronous modalities such as videoconferencing have appeared comparable, or better, in research contexts than face-to-face delivery in terms of quality of care, reliability of clinical assessments and treatment outcomes and adherence [10, 16–18]. The attitudes of clinicians who have experienced synchronous telemental health in research settings also appear to be largely positive, with professionals finding it both effective and acceptable [19]. There is also positive health economic evidence, with several studies suggesting telemental health is no

more expensive than face-to-face delivery and tends to be more cost-effective [17]. The use of remote consultations appears to be welcomed by many service users, who find it as satisfactory as face-to-face alternatives [2, 15, 20, 21], and in some cases preferable [22], even where they have initial reservations or limited experience of using computers [23]. However, studies assessing telemental health care tend to report small-scale and carefully planned interventions with volunteer participants, rather than large-scale emergency implementations, as in the current crisis [21].

The rapid, highly variable and often piecemeal adoption of telemental health modalities during the pandemic [7] has highlighted a range of very significant challenges, risks and implementation barriers. These include deterioration in the quality of care received by service users with certain mental health conditions; digital exclusion of those with limited technological access or expertise, or those facing significant social disadvantage, potentially exacerbating inequalities that already exist; lack of technological infrastructure and clear protocols within services; difficulty in establishing and maintaining therapeutic relationships; problems with conducting high quality assessments; and service users who lack private space or find participating in sometimes intimate and distressing discussions at home intrusive [2, 24–26]. A range of other ethical, regulatory, technological, cultural and organisational barriers to wider implementation of telemental health and its integration with routine face-to-face care have also been identified, both before and during the pandemic [17, 27–30].

Following the rapid shift to remote care during the pandemic, many policy makers, service planners, mental health professionals and service users have advocated for further evidence to inform the continued use of these technologies in the longer term [7, 31–33]. In order to achieve and sustain effective, integrated and acceptable implementation of telemental health approaches, and to identify the situations for which these are not appropriate, we need to understand how to optimise remote healthcare with a population that presents unique relational challenges associated with mental distress. This requires a nuanced understanding of the impacts of the rapid shift towards telemental health on clinical practice, quality and safety of care, equitable access to services, and the experiences of service users, carers and staff.

The current study focuses on the experiences of mental health service users already living with mental health conditions prior to the pandemic. While people already living with and receiving care for mental health problems have been active in writing personal accounts of the pandemic [2], few studies have systematically explored their experiences [26]. We report on a rapid collaborative framework analysis of service user interview data on views and experiences of telemental health during the pandemic, focusing especially on the following research questions:

a. What are the experiences, from a service user perspective, of the switch to telemental health care?

b. Which factors facilitate or impede people engaging with remote contact?

c. What are the advantages and disadvantages of specific remote care tools, in different setting and for different populations?

d. How do service users envisage the future for remote interactions in mental health services?

## Methods

Our study is a collaborative framework analysis [34, 35] of relevant material from interviews with people with pre-existing mental health conditions conducted as part of a broader study exploring loneliness and mental health during the COVID-19 pandemic [26].

Ethical approval was obtained from the UCL Research Ethics Committee on 19/12/2019 (ref: 15249/001) and an amended topic guide covering experiences of COVID-19, including telemental health, was approved on 14/08/2020. This paper reports findings from the second wave of interviews which took place in September-October 2020.

Both the data collection and the current analysis used a participatory, coproduction approach [36]. The research team included mental health service researchers, lived experience researchers (LERs) with personal experience of using mental health services and/or supporting others who do, and mental health clinicians, many with two or more of these roles. We conducted weekly team meetings to discuss the analysis methods and results, and collectively wrote this paper.

This work is part of the National Institute for Health Research Mental Health Policy Research Unit's programme in response to the COVID-19 pandemic, agreed in discussion with policy makers including Department of Health and Social Care and NHS England officials.

## Sampling and recruitment strategy

Initial recruitment took place between 7th May and 8th July 2020. We included adults living in the UK who identified as having had experiences of mental health difficulties that had begun prior to the pandemic. We recruited conducted via relevant community and voluntary sector organisations and networks, and via social media, with support from the Mental Elf blogger. In this paper we report findings from the second of two waves of interviews which was conducted in September-October 2020: this second interview included detailed questions about telemental health, as it was identified as a high priority area for data collection following discussions with policy makers and other stakeholders.

Purposive sampling, reflecting characteristics relevant to the research questions, guided participant selection (diagnosis, use of mental health services, gender, ethnicity, sexual orientation, and rural/urban setting). We reviewed our sample during recruitment and implemented targeted strategies to ensure diversity, such as approaching organisations with a specific remit to work with Black and Minority Ethnic communities and the use of targeted recruitment materials in an attempt to increase participation amongst these communities.

Potential participants were sent an information sheet and given the chance to ask questions, then formal consent was taken and recorded before the interview. Those who gave informed consent and participated in an initial interview were approached again approximately three months later and asked if they would like to take part in a follow-up interview. Participants gave audio-recorded verbal informed consent to being interviewed a second time.

## Data collection

Interviews were conducted by a team of LERs, including PS, PN and BC and KM (lived experience commentary authors, see Box 1), using videoconferencing or freephone options within the Microsoft Teams application. A second study researcher was present to support recording and save the interview in password protected university folders. Consent was audio-recorded before the interview. Interview audio files were then transcribed verbatim by a transcription company and all personal identifiers were removed for analysis.

Interviewers were generally paired with the same interviewee for the first and second wave of interviews. Other than this, there was no relationship between interviewers and interviewees prior to the interview and they corresponded only in order to arrange a time for the interview and answer any questions about the study. University researchers provided training for interviewers on the practicalities of conducting interviews on Microsoft Teams, obtaining verbal

Box 1. Lived experience commentary written by Beverley Chipp and Karen Machin on 24/01/2021

Our commentary encompasses observations from several Lived Experience Researchers. Common concerns were exclusion and choice. Person-centred care means everyone should be offered choices about whether to accept digital options, with shared decision-making, regularly reviewed without assumptions.

Interviews commenced at a point in the pandemic when there were few alternatives for support. Gratitude for any contact under the circumstances, and satisfaction within this context should not be used to justify any narrowing of choice in future provision.

The rapid adoption of digital shows how swiftly services can change, but any longer-term future policy should avoid making remote consultations a default option. We hear concerns that decisions are driven by budgetary, particularly estates, considerations, and that clients' needs may be secondary, with requirements to meet certain criteria before face-to-face options are offered. Hidden costs such as clinician time lost addressing technical challenges remain unaddressed.

Future research is urgently needed on safety and risks of video calls, particularly with group-work addressing self-injurious events impacting on others, and complexities of support. The quality of therapeutic relationships should also be considered. Choice, including how it is supported and encouraged, needs scrutiny. Hybrid approaches could be explored. The needs of people who are digitally excluded and consequently unable to take part in this study need to be understood. Initiatives should not exacerbate existing digital inequalities such as incompatibility with assistive technologies, poor Wi-Fi in residential facilities or inequalities considered under the Equalities Act.

Co-produced research can bring insights into emerging experiences from a grass-roots level. For example, community and peer support is under-represented in this paper, which focuses more closely on statutory services provision and existing literature emphasising 'psychotherapy' and 'treatment', whereas many people relied upon and provided support for their community over the pandemic. Survivor/service user led research may start with different questions ensuring studies are ahead of the curve at times when agile responses are required for unprecedented situations.

informed consent, dealing with distress and content of topic guide through a 2-hour online workshop. Additionally, a weekly peer-facilitated reflective space provided LERs with emotional support and space to discuss the research process.

Interviews were guided by a semi-structured interview schedule (see S1 Appendix). A previous study on loneliness, emerging literature on COVID-19 and mental health [2, 7], and lived experience within the research team informed the topic guide. It included a detailed exploration of experiences and views about remote contacts and questions relating to acceptability, preference (current and future), and differential experiences across modalities and clinical settings.

## Analysis

We conducted a two-stage collaborative framework analysis [34, 35]. This was an iterative approach with team discussions, including LERs, clinicians, and researchers at each stage to

enhance reflexivity and develop consensus on coding approaches and development of themes/topics.

In the first stage, interview data relevant to the research question was selectively extracted and indexed. We identified data from the interviews which referred to service users' experiences and perceptions of telemental health care, including data relating to:

a. Remote provision of support through video-calls, phone and any messaging means.

b. Support provided by, or facilitated by, any organisation (NHS, social services, voluntary sector or community organisation, excluding unmediated support such as apps).

c. Support provided by both paid staff and volunteers, including peers (but excluding entirely informal support provided, for example, by family and friends)

For this data extraction, a set of broad categories/codes relevant to telemental health care was developed through iterative team discussions. These categories were based on preliminary data screening, interviewers' familiarity with the data, and our and other groups' previous work on this topic. In this way, we generated a thematic coding framework which included deductively and inductively derived categories. A data extraction form was developed using Opinio [37] and an initial set of five interviews was coded to assess the specificity and adequate fit of the categories to the data. Based on this initial test, the coding book was refined through group discussion, and RA, PN, MS, PS, MB, UF, NL and LSR proceeded to index interview data under the pertinent categories. An "other" category was included to index relevant data that did not fit in any of the existing categories. The full list of categories is presented in Table 1.

At a second analysis stage, a smaller group of researchers (RA, PN, PS, MS and NV) reviewed the data collected within each category and synthesised the key emerging topics/themes. Emerging themes were identified through iterative cycles of group discussion and returning to the data, and related themes were grouped under headings/dimensions until a refined coding framework was developed. Codes were mapped across the data to report on variations between respondents. The results were then shared and discussed with the broader research team, giving each member a chance to reflect on key topics and assess whether the refined set of codes encompassed all the relevant data. Following this discussion, the smaller group of researchers further refined the results and selected quotes from interview transcripts.

## Results

### Sample

Of the forty-nine people who participated in the first wave of interviews and agreed to take part in a follow-up interview, one did not take part due to work/home demands, one preferred not to take part, and three could not be reached. Forty-four participants took part in the follow-up interviews analysed in this study. The interviews took place between 1st September and 14th October 2020. A majority of the sample was female (N = 28, 63%), aged between 26–55 years (N = 33, 75%), and living in urban settings (N = 35, 80%) across England. The main ethnic groups were White/White British (N = 28, 63%), Black/Black British (N = 6, 14%), and Asian/Asian British (N = 6, 14%). The majority (N = 33, 75%) reported current or recent mental health service use (this was during periods of lockdown or social distance regulations in the UK in 2020), mainly of NHS community mental health services (N = 22, 50%). Table 1 presents further characteristics of the participants.

We report emerging topics under four headings: (1) *Varying settings for telemental health*, (2) *What works for whom*: *experiences and preferences*, (3) *Patient safety and privacy*, and (4)

**Table 1. Characteristics of participants (n = 44).**

| Characteristic | Category | Number (%) |
|---|---|---|
| Gender | Female | 32 (73%) |
| | Male | 12 (27%) |
| Age | 18–25 | 3 (7%) |
| | 26–35 | 14 (32%) |
| | 36–45 | 13 (29%) |
| | 46–55 | 6 (14%) |
| | 56–65 | 3 (7%) |
| | 66–75 | 3 (7%) |
| | Information not available | 2 (4%) |
| Ethnicity | White British | 24 (54%) |
| | White Other | 4 (9%) |
| | Mixed/Multiple Ethnic Groups | 3 (7%) |
| | Asian/Asian British | 6 (14%) |
| | Black/Black British | 6 (14%) |
| | Other Ethnic Group | 1 (2%) |
| Sexuality | LGBTQI[1] | 8 (18%) |
| | Heterosexual | 30 (68%) |
| | Prefer not to answer or information not available | 6 (14%) |
| Region of UK | North (North East, North West, Yorkshire and Humber) | 8 (18%) |
| | Midlands (West Midlands, East Midlands) | 5 (11.5%) |
| | South (South East, South West and East of England) | 5 (11.5%) |
| | London | 25 (57%) |
| | Wales | 1 (2%) |
| Urban/Rural Location | City | 35 (80%) |
| | Town | 8 (18%) |
| | Village | 1 (2%) |
| Living Situation | Lives alone | 21 (48%) |
| | Lives with parent | 3 (7%) |
| | Lives with partner | 8 (18%) |
| | Lives with partner and child | 1 (2%) |
| | Lives with child | 2 (5%) |
| | Lives with parents and siblings | 3 (7%) |
| | Lives with partner and carer | 1 (2%) |
| | Lives with parent, partner and child and brother and family. | 1 (2%) |
| | Lives in shared accommodation | 3 (7%) |
| | Lives in mental health specific accommodation for homeless people | 1 (2%) |
| Self-Reported diagnosis | Personality Disorders | 6 (14%) |
| | Mood Disorders (Depression, Anxiety, PTSD) | 20 (45%) |
| | Bi-polar Disorder | 5 (11%) |
| | Schizophrenia/Psychosis | 6 (14%) |
| | Other (addictions, suicidal thoughts, OCD) | 3 (7%) |
| | Not stated | 10 (23%) |

*Note*. All characteristics were self-defined. LGBTQI included: Gay (n = 1), Lesbian (n = 1) Bisexual (n = 5) Pansexual (n = 1).

*Views about the future.* All topics were present to a greater or lesser extent across interviews with participants with different demographic characteristics; cases where a topic related to respondents' specific clinical or social circumstances are mentioned where applicable.

**1. What works for whom: Experiences and preferences.** Participants' preferences regarding remote care were dynamic and affected by the reason for contacting care providers, their relationship with the care provider, and both parties' ability and acceptability to use remote technology. Variations in people's preferences were dependent on individual inclinations that could vary over time, rather than being explained by participants' characteristics, circumstances or access to technology.

We identified two sub-themes within this topic: (1) *Remote care experiences and preferences*, and (2) *Barriers and facilitators to remote access*.

*Subtheme 1.1*: *Remote care experiences and preferences*. Participants' experiences of and preferences for remote mental health care varied greatly. While some participants welcomed remote therapy sessions, others reported that conversations often felt less engaging compared to face-to-face contacts.

> *Having counselling over the phone is very liberating and it's very freeing. I really, really enjoyed it. [P41: female, Black British, 36–45, city]*

> *[Video calls] because of my actual specific illness [. . .] it wasn't helpful. It was creating more distress. So, whilst the technology was brilliant, it was very distressing. [P30: female, White British, 36–45, town]*

Participants had varying preferences for the method of remote communication depending on the context and nature of clinical contact. Some preferred phone calls to video calls because they felt less intrusive, whilst others preferred video calls because they allowed both the service user and the therapist to pick up on visual cues. A few preferred email or text as they found written forms of communication easier and sometimes thought it helped with managing privacy issues.

> *I never saw the face of the person that I was chatting with, or even heard their voice, but we did build up a kind of professional relationship through messaging each other over the message service thing. And you could see the records of the conversation, which I think was helpful. [P31: female, White Asian, 18–25, town]*

Most favoured using text-based communication only for scheduling appointments or requesting medication repeat prescriptions. This indicated a distinction between 'functional' and 'relational' appointments: contacts to exchange practical information with less room for ambiguity (functional), vs. contacts to express states of mind and interact (relational).

> *With something like a therapeutic thing, there is so much that goes on- I don't know, with body language, eye contact, developing empathy, you know, being comfortable with silences, that doesn't work if you are using Zoom or something. [. . .] in a therapy type thing, there is something just very sort of special about being in that room. In a room like that, it has got to sort of. . . I mean, it sounds a bit ridiculous, and it is exaggerated to say sacred, but it is a very-it is a special thing. [P40: male, White British, 26–35, small town]*

> *Phone appointments are okay with the care co-ordinator, but not so brilliant with the psychiatrist, [. . .] because there's more to talk about with the psychiatrist, it's better to either see them in person or see them by video. . . [P25: female, African, 46–55, city]*

*I think there are a lot of things, like medication repeats where they haven't been set up as an automatic repeat or dosage changes, discussions. A lot of things like that, that I think could just be dealt with, with a telephone call. [P22: male, White British, 26–35, town]*

Not seeing staff's body language and other social cues generally made participants feel less connected and overall has a negative impact on participants' experiences of relational appointments. For online support groups run without live video, this led to reduced engagement and a lack of connection with other group members.

*Because of autism, I sometimes find it hard to focus on what people say, unless there is some sort of visual connection to it. [. . .] The ideal would be if everybody could just have subtitles built in. But the second best to that is being able to see somebody talking as they speak. [P27: female, White British, 26–35, city]*

Participants who started mental health support or changed their therapist/psychiatrist during the lockdown said that the relationships with the new care provider often felt less personal compared to face-to-face. This was particularly the case for initial assessments over the phone, as participants found it difficult to convey how distressed they were and felt that this affected the care they were offered afterwards. Conversely, familiarity with health providers facilitated remote contact.

*Since I have always managed to get to see the same GP. . . there is an established relationship. So it doesn't matter quite as much if I am not able to see him face to face. [P40: male, White British, 26–35, small town]*

*Subtheme 1.2*: *Barriers and facilitators to remote access*. Participants appreciated having access to remote care during the pandemic because it allowed them to continue with their mental health care when there was no other option. Some also wanted this mode of care to remain in the future, for example, participants who struggled to travel due to mobility issues or anxiety.

*Yes, it's [video calls] better for me, because I've got mobility issues, it means that I don't have to travel. So, sometimes. . . I would miss appointments because I just didn't have the get-up-and-go to leave the house, or I'd have a panic attack at the front door or something. [P9: female, other ethnicity, 36–45, big city]*

The switch to remote care facilitated access to different forms of support for some people who would have otherwise not been able to receive care, such as those admitted to mental health wards or participants living in remote areas that have only limited services available.

*I feel like it is only because of the fact that it is virtual now that the people can do this. A lot of members are signing on from hospital beds and retreats for people who are suicidal and things like that. [. . .] There was one member who was in a general hospital [. . .] It would not have been possible for him to come to the group had it been a face-to-face group. [P29: female, African, 26–35, city]*

*I've actually made more friends in the lockdown, through groups that I wouldn't have joined otherwise. (. . .) Online communities, and the group therapy is a local group. So I guess I've got more of a sense of community there. But I just literally wouldn't be able to go to all these groups if they weren't online. [P8: male, White British, 36–45, city]*

However, some participants did not have the necessary technology, internet connection, or private space to receive remote care. These factors increased the stress and difficulty of engaging with remote care, and in some cases resulted in users having to terminate their therapy sessions completely, or not being able to access remote care at all, leading to some participants feeling abandoned by services.

*The private therapist I see on Zoom, which is actually okay, but when I'm in the crisis house, the Wi-Fi there is non-existent. [P42: female, Asian, 18–25, city]*

*My cognitive analytical therapy ended up stopping because I couldn't find somewhere that felt private and safe to have those conversations, because obviously we couldn't do them face to face. [P19: female, White British, 36–45, city]*

*Most platforms [. . .] are completely incompatible with the disability [. . .] assistive technology that I use. Because I can't type using my fingers, I exclusively use dictation. [. . .] On online calls, like if I am on the usual Skype or Zoom and that, it doesn't allow me to write things with my voice without disconnecting the service. I can disconnect it and then write something and then copy and paste it subsequently, but I can't keep the call going whilst I am doing it because of the way the platforms are set up. [P44: female, White British, 36–45, town]*

Additionally, there were multiple examples of contexts in which people reported not being able to cope with technology, such as, when they were feeling severely unwell; experiencing paranoia and/or distress; and when there was lack of trust and interpersonal connection with service providers.

*Because with accessing healthcare I find the phone very ineffective, and it brings a lot of its own problems, when I get unwell I really can't cope with technology anyway. I will just wrap my phone in foil and put it in the garden when I am not feeling well, just get rid of it until a later date when I feel able to unwrap it. [P44: female, White British, 36–45, town]*

Access to services was particularly impeded when service providers were inflexible regarding the type of remote technology used, for example, insisting on using platforms only accessible with laptops or smartphones rather than phone calls.

*Some people don't have the technology and, you know, are missing out, and nobody calls them. [P48: female, White/Mixed, 46–55, city]*

Flexible service delivery and adaptability were key to reaching different populations and providing personalised care. Those who were not offered a choice regarding the delivery of care or the opportunity to have video-call voiced disappointment and perceived care provision as inequitable.

*So twice a week phone calls, for as long as we couldn't see each other, but if I chose not to have one of my sessions, she would add it onto what I get face to face at the end. And, to be honest, that made it—I mean, I appreciate her flexibility in that. [P1: female, White British, 36–45, city]*

*Again, it's like a postcode lottery, what area you live in [. . . influences] what support you get. To me, that's not okay. So, restricts people depending on what practice you use and what support you can get. [P4: female, White British, 26–35, city]*

Many participants reported not having received support or information to help them engage with remote care. This was an issue particularly for older service users who did not have the necessary devices or knowledge to use technological devices or were not offered the option to use services that they were familiar with.

*I wanted to get to use a computer [. . .]. And it said on the leaflet, "We run computer courses." So, I go along to the head office and it says, "No, we are not open. Go round the corner and walk half a mile and come to our other office." I walked all the way there and it said, "We are not open." [. . .] It has really put the kibosh on a lot of things that people would like to do. They just aren't really terribly available." [P13: male, White British, 66–75, London]*

*I have certainly been in that kind of place where I have just wanted to text somebody, but I have never seen them available for my age group, to be honest. I suppose maybe the younger generations when they feel suicidal maybe they hit the computer. I don't know. Maybe that is the thing for them, but maybe it might not be so for me. [P13: male, White British, 66–75, city]*

There were examples of service providers helping service users to access and adjust to remote care. For instance, some service users received information sheets and support from their therapist on how to use technology and switching to remote care. Other examples of service strategies to facilitate access to care were appointment reminders and charities promoting their services and reaching out to people.

*A week before your appointment, she would text you and say, "We're due to meet next week at this time." She did it off her own back. [. . .] I think that's really good that she does that because it shows she's thinking ahead, "I'm due to see you," and for people that don't remember- [. . .] that shows to me she's aware of who she's going to see, when they're coming and she's got you in mind. [P4: female, White British, 26–35, city]*

**2. Varied settings for telemental health.** Table 2 presents the range of remote care tools used by participants. These included video and phone consultations, phone helplines, text helplines, and email appointments. Table 3 presents the mental health care services that participants were in contact with, such as NHS community mental health services, inpatient services and private sector psychotherapy. Some participants described experiencing difficulties to stay connected with community mental health services.

*The community mental health team that I've been under haven't been brilliant over lockdown. I obviously know we can't see each other in person, but it's literally gone completely quiet for most of the six months [. . .]. I'm supposed to hear from my care co-ordinator every*

**Table 2. Modalities of care experienced by participants (N = 44).**

| Modality | Number of participants |
| --- | --- |
| Individual videocall (therapy and other appointments) | 13 (29%) |
| Phone contacts (therapy and other appointments) | 36 (82%) |
| Phone helplines | 10 (23%) |
| Text-based consultations (email and other messaging for therapy and others) | 21 (48%) |
| Text helplines | 4 (9%) |
| Online support groups | 14 (32%) |
| Recovery Colleges | 3 (7%) |

**Table 3. Mental health services accessed by participants.**

| Service | Number of participants |
|---|---|
| None or on waiting list | 11 (25%) |
| NHS Community mental health services[2] | 22 (50%) |
| Inpatient Services[3] | 3 (7%) |
| GP or Primary Care counselling | 8 (18%) |
| Private sector psychotherapy only | 1 (2%) |
| Voluntary sector mental health services only. | 2 (5%) |

*Note.* Community Mental health services included: Community Mental Health Team (n = 16) Reablement team (n = 1) Therapist (n = 5) NHS Peer Support Service (n = 1) 3. Inpatient services included: Acute inpatient ward (n = 1) and Crisis house (n = 2). Services were accessed in 2020, coinciding with periods of lockdown or social distancing in the UK.

> week and I guess they're busy [. . .]. So, I literally have heard nothing, pretty much [. . .]. . . over lockdown I've probably heard from them [by phone] about two or three times. [P25: female, African, 46–55, city]

Many participants relied on services other than secondary mental health services, such as their General Practitioner (GP), charities, crisis helplines and/or online support groups which offered a variety of new forms of remote contact that were seen as broadening choice and service availability. Overall, participants reported having received high quality care, information, and resources from these service providers and identified them as providing valuable support.

> I actually think charities did a great job. (. . .) I felt like there was a lot of content and a lot of, like, helpful hints, and tips, and guides, and suggestions, and recommendations, and resources available. [. . .] I think charities did a particularly good job at pulling together a lot of information that could be disseminated, and people could pick and choose what they wanted to digest. [P21: female, White, 36–45, big city]

Sustained contact via remote methods with GPs was particularly valued due to the possibility of keeping medication prescriptions up to date and assessing potential urgent need for physical health care. The reliability of GP services helped reduce the impact of worries about physical health on participants' mental health. Overall, interviewees welcomed the efficiency and support provided by GP surgeries during the pandemic and reported relying on this support when specialised health services were reduced or inaccessible. GPs' mental health support and active listening were very important in reducing participants' anxiety about potentially losing access to services. Examples of helpful GP strategies for providing remote care were scheduling weekly check-ups regardless of symptoms and offering email consultations.

> From GP I have done email consultation online form and then he emails me back. (. . .) It feels much more efficient at times. Like, just get to the point. And in some respects, I feel like be more honest like that. (. . .) the reception option is a bit scary (. . .) [I want to] avoid them. [P15: female, White British, age not specified, town]

Some participants saw having to register on apps and fill out complicated online forms as an important barrier to accessing this care.

*Normally, if I need to speak to a GP, it's much easier to get an appointment. And these days it's quite hard and they resist even giving you a phone appointment because they prefer you to do this eConsult form where you basically fill out a list of answers to some questions online, and then somebody might text you back (. . .) But it's harder to actually get to speak to anyone.* [P1: *female, White British, 36–45, city*]

Participants appreciated having the support of charities, voluntary sector organisations and crisis helplines. However, some participants perceived crisis helplines as dismissive, and call handlers less knowledgeable than mental health staff about mental health. One person additionally stated that NHS crisis helplines did not offer support in different languages despite being promoted in several languages. Needs not being met by helplines which were thought to be available 24/7 was perceived as a risk to personal safety.

*Sometimes I just needed somebody to talk to, and [the helplines] were always available. I think the [helpline] have helped me the most. Because they just listen and understand. I'm not in a crisis today, but if I was having a bad day, it would be very useful to phone the [helpline].* [P18: *female, British Asian, 36–45, city*]

*Not the mental health lines [. . .]. People are not well trained, just causing more distress than anything else.* [P48: *female, White/mixed, 46–55, city*]

**3. Patient safety and privacy.**   New situations occurring in the online space, such as inpatients joining community peer support groups described in *Subtheme 2.2*, had safety and privacy implications that had not been encountered in face-to-face care.

Some people described feeling unsafe during remote consultations either due to a lack of privacy and safety at home, and this sometimes led to a pause in treatment. Text-based services were cited as being a helpful alternative where safety or privacy at home was compromised.

*My cognitive analytical therapy ended up stopping because I couldn't find somewhere that felt private and safe to have those conversations. . .Basically, if I was upstairs my boyfriend could hear me because he is working from home [P19: female, White British, 36–45, city].*

*[Remote therapy via text is better than via phone] for people that can't talk because it is not safe to. . .You can delete the texts if you are in that kind of situation, where you don't want people to know what you are saying or you are worried about your privacy. I would love to see more text-based support in the UK.* [P19: *female, White British, 36–45, city*]

There were also reports of people feeling that non-verbal signals of escalating distress and agitation were being missed by clinicians, particularly over the telephone, potentially leading to safety concerns.

*There was one incident in particular where she, towards the end of the session, said something that really upset me. And she didn't realise that I was upset because she couldn't see me. And I did try to, kind of, tell her and I thought she had heard it in my voice that I wasn't okay. [. . .] But I don't think she had any idea [. . .] And actually later that evening, I ended up in A&E because I was quite upset and not really coping with things, [. . .] I felt like that might have been different had we been face to face. . .* [P1: *female, White British, 36–45, city*]

**4. Views about the future.**   Several participants identified aspects of telemental health that they would like to see incorporated into their care in the future. Their comments suggested

that a hybrid model of care delivery could combine the advantages of face-to-face care, including developing the therapeutic relationship, with the advantages of remote care, such as flexibility and reduced need for travel.

> *If I was given a choice then I would probably say that I would want some of my appointments to be face-to-face, some of them to be virtual, and the remaining can be done over the telephone. [P49: male, British Asian, 46–55, city]*

> *I think the first couple of times I'm meeting s different had we been face-to-face [P1: female, White British, 36–45, city]*

Some participants reported incidents of severe distress during or after remote sessions. For example, a participant described risky or self-harming behaviours taking place during a video support group. This had an impact on all members of the group and new ground rules, risk management and support were put in place in subsequent sessions. Another participant raised concerns around attending online group therapy sessions where they were not allowed to have their cameras on and how this had potential implications for safety.

> *Something big that happened in one of the groups that I don't think would have happened in a face-to-face group. There was this one guy, a member, [self-harmed] on camera. I think it is unlikely that he would have done that in a face-to-face group. . .an ambulance was called straight away. . . And it was quite difficult to watch, actually, because I have got personal experience of [self harm]. [P29: female, African, 26–25, city]*

A blend of face-to-face and digital support groups was also identified as beneficial to balance the advantages of both methods, with face-to-face perceived as facilitating stronger interpersonal connections, whereas online groups provided greater flexibility and anonymity, which was sometimes preferred. A participant explained that her online therapy group worked well because they had already met face-to-face before the pandemic. Conversely, a participant who started attending a new online group where no-one knew each other felt there was very poor engagement.

Service users also identified other scenarios in which a blended approach would be beneficial, for example, receiving phone calls when on a waiting list for face-to-face therapy, or for "functional" appointments as described in *Subtheme 1.1*.

> *A real-life appointment with PPE [Personal Protective Equipment] is massively preferable to a phone consultation. . . Unless it is to do with something purely factual like [. . .] explaining something to do with medication or something. Then, in which case, certain circumstances a phone call is fine. [P40: male, White British, 26–35, small town]*

Overall, however, many participants expressed a preference to return to face-to-face for relational appointments such as psychological therapy in the future, as technology is *"not really a replacement for the real thing" [P3, female, White British, 36–45, city],* but were happy to use remote contact methods for more functional appointments such as brief medication reviews with their GP or psychiatrist (relational and functional appointments are described in Subtheme 2.1).

> *I don't think most of the appointments, especially with things like mental health medication where you have to speak directly to a GP while you're still increasing dosages, making sure you're tolerating them properly and they won't give you just a repeat prescription for some*

*time as you're starting a new medication. That sort of follow up appointment can definitely be
done just over the telephone.* [P22: male, White British, 26–35, town]

## Discussion

### Main findings

We analysed data about telemental health care from interviews with forty-four service users
who experienced mental health problems with onset prior to the COVID-19 pandemic. A dia-
gram mapping key findings is presented in Fig 1. Service users appreciated remote care options
during the height of the pandemic when other forms of care were not possible. However,
remote care was mainly seen as an option to allow access to care in extreme circumstances,
rather than an alternative of comparable quality to face-to-face care.

We found that participants' preferences for modality of care varied between participants and
within participants. Factors that influenced preferences included contextual factors, such as having
a private space for therapy, and individual factors, such as attitudes to technology, mediated human
contact. Differences were identified between relational appointments such as those to participate in
psychological therapy, and functional appointments to renew medication prescriptions or complete
quick health checks. Other variables influencing remote care experiences were the relationship with
the care provider, including whether they had met face-to-face in the past, and ease of use or access
to necessary technology. Overall, participants stressed the need to provide alternatives for people
who could not access or did not feel comfortable with telemental health care.

Strategies used by GP practices and charities, such as scheduling regular phone calls and
signposting to available resources, were mentioned as measures that facilitated remote care.
Some new risks and challenges were also identified for example in online group sessions or in
initial mental health assessments conducted over the phone.

### Relation of findings to previous literature

We found that service users valued personalised, flexible options that include a combination of
different types of remote and face-to-face contact. These preferences are in line with reports

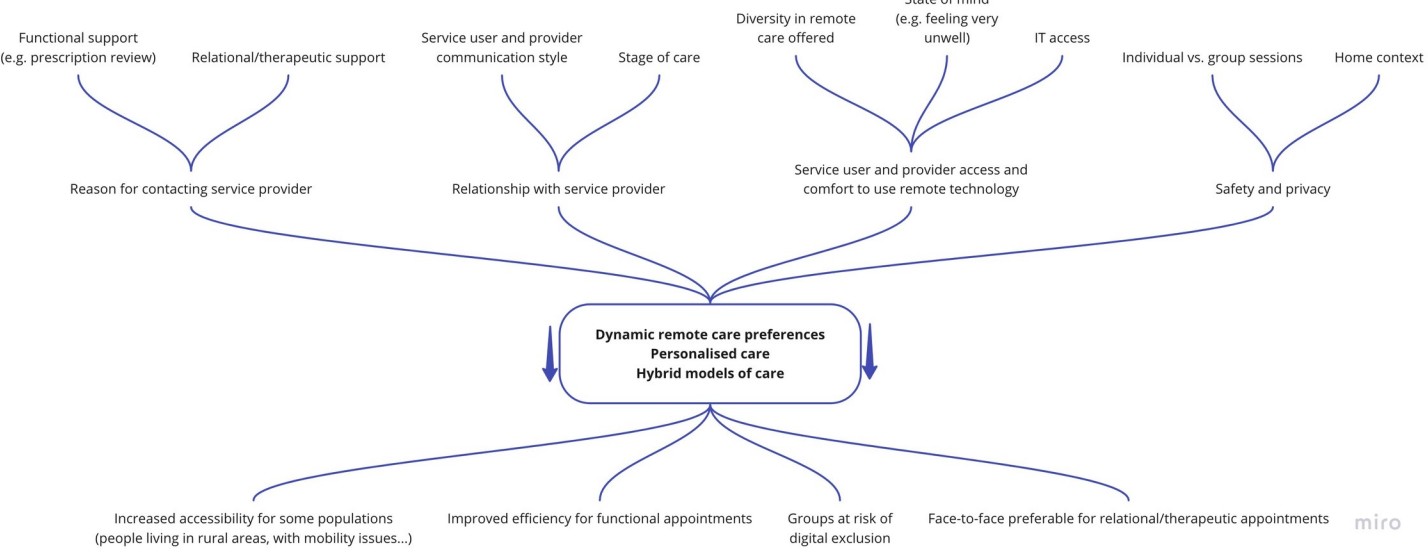

**Fig 1. Diagram mapping key findings.**

prior to the pandemic indicating people's desire for mental health services that offer choice and responsiveness to individualised needs and circumstances [38]. It was apparent in our study that experiences and preferences regarding telemental health were essentially personal. Acceptability varied greatly between participants in ways that could not be readily explained by their situation, thus suggesting it is not possible to make assumptions about participants' preferences.

In line with previous research, participants mentioned beneficial aspects of telemental health care and innovative strategies that services adopted to address safety and technology challenges during lockdown [22, 23, 39, 40]. These include improved accessibility and continuity of care, especially where difficulties such as physical mobility, social anxiety or paranoia impede travel and face-to-face contacts, increased convenience for those facing geographical barriers, convenience and communication within and between mental health teams [2, 7, 16, 21, 23, 40]. However, participants also commented on the risk of digital exclusion of those lacking the necessary skills, resources, or privacy to engage with remote services. This is particularly notable given that our participants were sufficiently digitally connected to be able to take part in the study. Digital exclusion appears not to have been widely addressed in previous studies [2, 7]. Groups who were already disadvantaged, such as older adults, people with sensory or cognitive impairment and minoritised groups are most at risk [41–44]. An example are people experiencing significant social disadvantage or severe mental health problems including psychosis have been reported to benefit less from telemental health [45]. Evidence is lacking both on the extent of digital exclusion and on how it might be overcome [21]. Barriers to access care such as those described in *Subtheme 1.2* may help to explain why, despite its robust research evidence base and the strategic focus in England on more effective integration of digital technologies across the NHS [46], implementation of telemental health had remained very limited prior to the pandemic in England and in other countries with similar mental health systems [1].

Participants highlighted limitations of remote care that went beyond lack of access. Service users commented on different contexts in which remote care seemed more or less appropriate. Previous literature has varied in reports about how far video-calls can offer an authentic substitute for the connection made between service user and professional face-to-face. Regarding relational appointments, research involving female older adults [23] and veterans [22] has reported that service users tend to find video more impersonal than face-to-face due to reduced physical cues, and feel more comfortable talking to therapists in person, where possible. Conversely, a systematic review found that in terms of therapeutic alliance, only a minority of studies reported video-based interventions as inferior to face-to-face treatment [39]. Our study suggested people vary in the extent to which they feel rapport and therapeutic alliance can be of equivalent quality to face-to-face, but most seemed to feel it was to some extent inferior: it may be that the volunteer research participants in previous studies have been particularly open to seeing digital contacts as equivalent, or that better planning and preparation has improved experiences in these previous research studies. As with the present study, the evidence is mixed and potential negative impacts on rapport and therapeutic relationships, leading to more superficial therapeutic contacts, have been noted, including during the pandemic [2].

Many felt that remote care tools were inherently a less satisfactory way to form a therapeutic relationship, and some discarded remote care options altogether. However, we found potentially remediable barriers to delivering good quality mental health, such as, the service provider's ability and level of comfort to use technology. Specific barriers to engaging with remote care identified in our interviews and previous literature were lack of familiarity with and mistrust of relevant technology, low image quality on video calls, connectivity problems, and

audio delays [22, 23]. These concerns have been shared by mental health service staff in the UK [7] and professionals have tended to report a preference for face-to-face contact for both assessment and treatment [21]. Privacy and safety have also been emphasised in the literature as serious issues to consider in telemental healthcare provision [21, 24, 47, 48]. Research has suggested that telemental health is potentially effective for group interventions and our participants described benefits from newly established online groups, [12], but the distressing situations described by some suggest a the need to develop guidelines to ensure service users' wellbeing and privacy during remote group care provision.

While face-to-face care remains preferable in some situations, participants also identified strategies adopted by service providers to facilitate engaging with remote care. This suggests there is scope for extending digital access for those who wish to receive it [21] and develop strategies to prevent patients with limited access from being at a disadvantage [49, 50].

## Strengths and limitations

Because of the pandemic context, recruitment for this study was mainly undertaken through social media, and interviews were conducted via Microsoft Teams. Although efforts were made to include participants who felt less comfortable with remote communication (recruitment via voluntary sector and community groups, providing a phone interview option), our paper may under-report problems with remote working. At the same time, remote data collection allowed us to reach a sample across England and attending a range of mental health services.

Our findings provide a snapshot of a specific short period at a stage in the COVID-19 pandemic when social distancing restrictions were in place to varying degrees in England: some aspects of service provision and service user views and experiences may reflect this particular time and may have subsequently changed. However, the significant contributions of LERs and clinicians ground this research in real-world experiences and increases the applicability of our findings. An important strength of this study is the collaborative nature of the analysis, combining larger multidisciplinary group input with smaller group consolidation of findings, resulting in a detailed rapid analysis. Our group's experience and previous work contributed to the identification of important issues, while leaving room for the development of inductively derived themes from the data.

## Implications

Remote care has been an important strategy to allow care to continue at a time of social distancing and is likely to remain a major modality for delivering continuing care in any similar emergency. There are also potential benefits in continuing some use of telemental health for future service delivery during recovery from the pandemic and beyond, including greater convenience and accessibility for some service users, as well as efficient service delivery. Our findings suggest that continuing some use of telemental health beyond the pandemic is feasible and acceptable from a service user perspective, but that further steps need to be taken to ensure that this is safe, high quality and in keeping with service users' individual needs. Guidelines equivalent to those in place for face-to-face care need to be developed to protect service users' wellbeing and privacy. At the same time, we found service users considered remote care was not an acceptable option in some situations and face-to-face care was required.

We found that that how people experience remote care and what they find acceptable is very individual: modality of care offered should thus ideally be discussed on a case-to-case basis to find the best fit for service users' preferences and circumstances. This requires flexible and personalised hybrid models of care take advantage of positives of telemental health care, as

well as offering face-to-face care when necessary or preferred: a mixture of types of contact may work well for many. Guidance around care options and access to telemental health care, such as those we found were provided by some charities and GP practices, should be further developed and routinely offered to service users in all mental health services.

A shift to remote care also has organisational, training and technological implications, such as the need for recruitment of IT support staff and development of policies and technology to allow for staff to adopt flexible approaches. Our research suggests staff urgently require training in assessing remotely what is likely to work best for each individual service user and overcoming communication barriers associated with technology, culture and language. Different approaches for service providers to identify and deliver the most beneficial packages of hybrid models of care should be assessed. Promoting communication between staff and service users will reduce assumptions and anxieties about new ways of working.

Research emerging from other countries has identified similar advantages and limitations to remote care, such as increased accessibility for groups who cannot travel, or difficulties resulting from not picking up on non-verbal cues during phone consultations [2]. This suggests our findings are also relevant to contexts outside of the UK. However, lack of health insurance coverage for remote care [51], or lack of access to medication prescribed during online appointments [52] are examples of context-specific challenges that were not identified in our sample. Further research is required into overall uptake of telemental health, the barriers and facilitators to engaging with it, and the unintended consequences and risks of exacerbating existing inequalities that may result from its use. We found relatively little evidence of innovative strategies to improve acceptability and reduce digital exclusion: potential approaches that may warrant further investigation include providing training in technology use and access to devices for service users. Questions for further research include impacts on therapeutic alliance and communication of telemental health adoption, and how best to implement telemental health in the longer term, including investigating the training needed by staff to become confident in using remote technologies and incorporating them in safe and flexible care pathways.

## Supporting information

**S1 Checklist. COREQ (COnsolidated criteria for REporting Qualitative research) checklist.**
(PDF)

**S1 Appendix. Follow-up interview topic guide.**
(DOCX)

## Author Contributions

**Conceptualization:** Norha Vera San Juan, Prisha Shah, Merle Schlief, Una Foye, Brynmor Lloyd-Evans, Alan Simpson, Sonia Johnson.

**Data curation:** Norha Vera San Juan, Prisha Shah, Merle Schlief, Rebecca Appleton, Patrick Nyikavaranda, Mary Birken, Una Foye, Brynmor Lloyd-Evans, Alan Simpson, Natasha Lyons, Luke Sheridan Rains, Sonia Johnson.

**Formal analysis:** Norha Vera San Juan, Prisha Shah, Merle Schlief, Rebecca Appleton, Patrick Nyikavaranda, Mary Birken, Una Foye, Nicola Morant, Alan Simpson, Natasha Lyons, Luke Sheridan Rains, Sonia Johnson.

**Funding acquisition:** Alan Simpson, Sonia Johnson.

**Investigation:** Alan Simpson, Sonia Johnson.

**Methodology:** Norha Vera San Juan, Una Foye, Nicola Morant, Alan Simpson, Sonia Johnson.

**Project administration:** Norha Vera San Juan, Brynmor Lloyd-Evans, Justin J. Needle, Alan Simpson, Zainab Dedat, Sonia Johnson.

**Resources:** Alan Simpson, Sonia Johnson.

**Software:** Zainab Dedat.

**Supervision:** Norha Vera San Juan, Brynmor Lloyd-Evans, Sonia Johnson.

**Validation:** Nicola Morant, Justin J. Needle, Alan Simpson, Luke Sheridan Rains, Zainab Dedat, Sonia Johnson.

**Visualization:** Norha Vera San Juan, Sonia Johnson.

**Writing – original draft:** Norha Vera San Juan, Prisha Shah, Merle Schlief, Rebecca Appleton, Patrick Nyikavaranda, Mary Birken, Una Foye, Brynmor Lloyd-Evans, Nicola Morant, Justin J. Needle, Alan Simpson, Natasha Lyons, Luke Sheridan Rains, Sonia Johnson.

**Writing – review & editing:** Norha Vera San Juan, Prisha Shah, Merle Schlief, Rebecca Appleton, Patrick Nyikavaranda, Mary Birken, Una Foye, Brynmor Lloyd-Evans, Nicola Morant, Justin J. Needle, Alan Simpson, Natasha Lyons, Luke Sheridan Rains, Zainab Dedat, Sonia Johnson.

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
