## [Decision Letter · Decision Letter 0]

24 May 2021

PONE-D-21-05340

Service user experiences and views regarding telemental health during the COVID-19 pandemic: a co-produced framework analysis

PLOS ONE

Dear Dr. Vera San Juan,

Thank you for submitting your manuscript to PLOS ONE. After careful consideration, we feel that it has merit but does not fully meet PLOS ONE’s publication criteria as it currently stands. Therefore, we invite you to submit a revised version of the manuscript that addresses the points raised during the review process.

Please pay careful attention to the length of the manuscript, the comments about the tables and clarifying some of the context-specific terms for mental health services.  It is also a journal requirement to adhere to standard reporting guidelines for qualitative research.  i suggest adding a COREQ or similar checklist as a supplementary file.

We look forward to receiving your revised manuscript.

Kind regards,

Bronwyn Myers

Academic Editor

PLOS ONE

Journal Requirements:

3. Please ensure that you refer to Figure 1 in your text as, if accepted, production will need this reference to link the reader to the figure.

Reviewers' comments:

Reviewer's Responses to Questions

**Comments to the Author**

1. Is the manuscript technically sound, and do the data support the conclusions?

Reviewer #1: Yes

Reviewer #2: Yes

2. Has the statistical analysis been performed appropriately and rigorously? 

Reviewer #1: N/A

Reviewer #2: N/A

3. Have the authors made all data underlying the findings in their manuscript fully available?

Reviewer #1: Yes

Reviewer #2: No

4. Is the manuscript presented in an intelligible fashion and written in standard English?

Reviewer #1: Yes

Reviewer #2: Yes

5. Review Comments to the Author

Reviewer #1: Thank you for your article. It was a great pleasure to review it. The article addresses a crucial and actual topic. The participatory methodology utilized is exciting and enhances the research objectives.The results are thoroughly described, the discussion addresses the proposed questions and the results and the implications suggested are meaningful. I recomend to publish the article as it stands.

Reviewer #2: This is an excellent manuscript that is important for the field as we consider new ways of working. i have some recommendations to strengthen it further:

1. There are some minor grammatical errors in the abstract that should be attended to

2. Please specify number of interviews in abstract

3. Background: greater attention to context of COVID-19 in the UK and how this impacted on care delivery is needed so that readers outside the UK can better understand the need for these service changes

4. In the methods please report according to COREQ or similar standard reporting guidelines for qualitative research- this is a journal requirement

5. How was consent for the interviews managed: remotely or in person

6. Table 1- some of the categories are hard to follow for an international reader- the urban/rural distinction is not clear for example. Would a town be urban or rural?

Table 2- please add %

7. At what point in the pandemic did service users access the services and organisations described in Table 2 and 3. I may have missed it, but I think it may be useful to foreground more at what point in the pandemic service users were interviewed and the time period in the pandemic that they were asked to reflect on. Their experiences of mental health services may have been very different in a "hard" lockdown compared to a softer lockdown or between waves of infection. This links back to point 3.

8. The results are very long as there is room to consolidate these by removing some of the quotes, especially where there is repetition.

9. Discussion. Any thoughts about whether these findings are relevant for settings outside of the UK?

6. PLOS authors have the option to publish the peer review history of their article (what does this mean?). If published, this will include your full peer review and any attached files.

Reviewer #1: No

Reviewer #2: No

---

## [Author Response · Author response to Decision Letter 0]

3 Aug 2021

Dear Prof. Myers,

We would like to thank you for sending us the comments made by the reviewers on our manuscript

Service user experiences and views regarding telemental health during the COVID-19 pandemic: a co- produced framework analysis as these have strengthened the manuscript. We include a point-by- point reviewer comments and our response to their feedback below.

REVIEWER 1

▪ Thank you for your article. It was a great pleasure to review it. The article addresses a crucial and actual topic. The participatory methodology utilized is exciting and enhances the research objectives. The results are thoroughly described, the discussion addresses the proposed questions and the results, and the implications suggested are meaningful. I recommend to publish the article as it stands.

Thank you very much for your positive feedback.

REVIEWER 2

Thank you for your recommendations, we have addressed each one and outline below:

▪ There are some minor grammatical errors in the abstract that should be attended to 

We thoroughly reviewed the abstract and have made some grammatical changes.

▪ Please specify number of interviews in abstract

We added this to the findings section of the abstract.

▪ Background: greater attention to context of COVID-19 in the UK and how this impacted on care delivery is needed so that readers outside the UK can better understand the need for these service changes

We added some information about key dates and changes experienced in the UK in the introduction page 3.

▪ In the methods please report according to COREQ or similar standard reporting guidelines for qualitative research- this is a journal requirement

We have completed a COREQ checklist to reflect compliance and attach it as supporting material.

▪ How was consent for the interviews managed: remotely or in person

Consent was obtained audio recorded remotely (specified now on page 5).

▪ Table 1- some of the categories are hard to follow for an international reader- the urban/rural distinction is not clear for example. Would a town be urban or rural?

Categories were self-defined. We have added a note to reflect this below Table 1.

▪ Table 2- please add %

Added now.

▪ At what point in the pandemic did service users access the services and organisations described in Table 2 and 3. I may have missed it, but I think it may be useful to foreground more at what point in the pandemic service users were interviewed and the time period in the pandemic that they were asked to reflect on. Their experiences of mental health services may have been very different in a "hard" lockdown compared to a softer lockdown or between waves of infection. This links back to point 3.

Services were accessed during periods of lockdown or social distance regulations in the UK in 2020, this is now mentioned in page 6 and under Table 3. We mention the potential differences in perceptions at different timepoints in in the pandemic in the limitations page 18.

▪ The results are very long as there is room to consolidate these by removing some of the quotes, especially where there is repetition.

We shortened or removed some of the quotes to make the results more concise.

▪ Discussion. Any thoughts about whether these findings are relevant for settings outside of the UK? We have added some reflections around this in the discussion page 19.

---

## [Editor Report · Decision Letter 1]

31 Aug 2021

Service user experiences and views regarding telemental health during the COVID-19 pandemic: a co-produced framework analysis

PONE-D-21-05340R1

Dear Dr. Vera San Juan,

We’re pleased to inform you that your manuscript has been judged scientifically suitable for publication and will be formally accepted for publication once it meets all outstanding technical requirements.

Kind regards,

Frédéric Denis, Ph.D.

Academic Editor

PLOS ONE
---

## [Editor Report · Acceptance letter]

8 Sep 2021

PONE-D-21-05340R1 

Service user experiences and views regarding telemental health during the COVID-19 pandemic: a co-produced framework analysis 

Dear Dr. Vera San Juan:

I'm pleased to inform you that your manuscript has been deemed suitable for publication in PLOS ONE. Congratulations! Your manuscript is now with our production department. 

Kind regards, 

on behalf of

Dr. Frédéric Denis 

Academic Editor

PLOS ONE